Sensor-based systems for the measurement of Functional Reach Test results: a systematic review

Francisco Luís 1
Duarte João 1
Godinho António Nunes 2
Zdravevski Eftim 3
Albuquerque Carlos 4 5 6
Pires Ivan Miguel 7
Coelho Paulo Jorge paulo.coelho@ipleiria.pt 1 8
1 School of Technology and Management, Polytechnic University of Leiria , Leiria , Portugal
2 Coimbra Institute of Engineering, Polytechnic of Coimbra , Coimbra , Portugal
3 Faculty of Computer Science and Engineering, University of Sts. Cyril and Methodius , Skopje , North Macedonia
4 Child Studies Research Center (CIEC), University of Minho , Braga , Portugal
5 Higher School of Health, Polytechnic Institute of Viseu , Viseu , Portugal
6 Nursing School of Coimbra (ESEnfC), Health Sciences Research Unit: Nursing (UICISA: E) , Coimbra , Portugal
7 Instituto de Telecomunicações, Escola Superior de Tecnologia e Gestão de Águeda, Universidade de Aveiro , Águeda , Portugal
8 Institute for Systems Engineering and Computers at Coimbra (INESC Coimbra) , Coimbra , Portugal
Zhao Wenbing
Electronic publication date: 2024 Mar 15
Publication date: 2024
Volume: 10
Electronic Location ID: e1823
Received 2023 Jul 14; Accepted 2023 Dec 26
Copyright: ©2024 Francisco et al.
Copyright year: 2024
Copyright holder: Francisco et al.
License: This is an open access article distributed under the terms of the Creative Commons Attribution License, which permits unrestricted use, distribution, reproduction and adaptation in any medium and for any purpose provided that it is properly attributed. For attribution, the original author(s), title, publication source (PeerJ Computer Science) and either DOI or URL of the article must be cited.
License URL: https://creativecommons.org/licenses/by/4.0/

Keywords: Functional Reach Test, Sensors, Technological devices, Physical diseases, Systematic Review, Ambient Assisted Living

Funding: FCT/MEC through national funds FEDER-PT2020 partnership agreement UIDB/50008/2020 National Funds through the FCT - Foundation for Science and Technology, I.P. UIDB/00742/2020 This work is funded by FCT/MEC through national funds and, when applicable, co-funded by the FEDER-PT2020 partnership agreement under the project UIDB/50008/2020. This work is also funded by FCT/MEC through national funds and co-funded by FEDER –PT2020 partnership agreement under the project UIDB/00308/2020 (DOI 10.54499/UIDB/00308/2020). This work is also funded by National Funds through the FCT - Foundation for Science and Technology, I.P., within the scope of the project UIDB/00742/2020. The funders had no role in study design, data collection and analysis, decision to publish, or preparation of the manuscript.

==============================
The measurement of Functional Reach Test (FRT) is a widely used assessment tool in various fields, including physical therapy, rehabilitation, and geriatrics. This test evaluates a person’s balance, mobility, and functional ability to reach forward while maintaining stability. Recently, there has been a growing interest in utilizing sensor-based systems to objectively and accurately measure FRT results. This systematic review was performed in various scientific databases or publishers, including PubMed Central, IEEE Explore, Elsevier, Springer, the Multidisciplinary Digital Publishing Institute (MDPI), and the Association for Computing Machinery (ACM), and considered studies published between January 2017 and October 2022, related to methods for the automation of the measurement of the Functional Reach Test variables and results with sensors. Camera-based devices and motion-based sensors are used for Functional Reach Tests, with statistical models extracting meaningful information. Sensor-based systems offer several advantages over traditional manual measurement techniques, as they can provide objective and precise measurements of the reach distance, quantify postural sway, and capture additional parameters related to the movement.

Introduction

Balance issues are common in older adults and individuals with neurological conditions, including multiple sclerosis and Parkinson’s disease (Park, Kang & Horak, 2015; Cuevas-Trisan, 2019; Norbye, Midgard & Thrane, 2020; Gheitasi et al., 2021; Gaspar & Lapão, 2021). A prominent clinical evaluation method to gauge a person’s balance and fall risk is the Functional Reach Test (FRT). The test measures the maximum distance a person can reach forward while maintaining a standing position without taking a step (Duncan et al., 1990; Hannan et al., 2021). The Timed Up and Go (TUG) test (Kear, Guck & McGaha, 2017; Soto-Varela et al., 2020) and the Berg Balance Scale (BBS) (Blum & Korner-Bitensky, 2008) are two other comparative tests used to assess different balance elements. The FRT concentrates on the subject’s capacity to stay stable while reaching forward. By using these tests, healthcare providers may assess a patient’s balance and fall risk and then create a customized treatment plan to assist the patient in regaining their balance and reducing their risk of falling. Exercise, physical therapy, and alterations of a person’s lifestyle can all help improve the balance (Halvarsson, Dohrn & Ståhle, 2015; Papalia et al., 2020). These assessments can aid in the early identification of balance problems and initiating appropriate therapies to avoid falls and enhance the overall quality of life (Bjerk et al., 2017). Mobile devices, including smartphones and tablets, commonly have a variety of sensors, such as magnetometers, gyroscopes, barometers, proximity sensors, ambient light sensors, GPS sensors, and accelerometers, which can aid in the automatic detection of different parameters (Oniani et al., 2019). Recently, some authors have suggested using various technologies to estimate the FRT. These include video and motion capture systems or console controllers that contain some sensors, such as an accelerometer, gyroscope, and magnetometer (Gomes et al., 2018; Springer, Friedman & Ohry, 2018; Pires, Garcia & Zdravevski, 2020; Gheitasi et al., 2021; Adıguzel & Elbasan, 2022). The research also included identifying mobile devices and data from their sensors to evaluate, monitor, and follow people’s clinical status and predict adequate therapy (Pires, Garcia & Zdravevski, 2020).

Integrating sensors with the FRT mainly aims to improve balance assessment and fall risk among different populations. We aim to gain more precise, objective, and real-time data to evaluate balance better. The research challenges include determining sensor accuracy and precision, identifying the best ways to integrate sensors into clinical practice without disrupting the FRT, developing data interpretation methods, ensuring user-friendliness, evaluating cost-effectiveness, assessing patient compliance, examining population specificity, real-world application, predictive value, and addressing technological challenges. These problems improve the objective assessment of balance and fall risk, which is crucial for preventing falls and injuries, especially in vulnerable populations like older adults. Although the FRT may utilize sensors to determine distance, there is currently insufficient data on these sensors’ precision and dependability.

Further studies are required to validate the accuracy and dependability of sensors in various populations and environments. As sensor technology advances, new sensors and models must be integrated and tested via ongoing studies. Data analysis tools must be devised to comprehend the enormous amount of data gathered during the FRT test. Clinical applications and longitudinal investigations are also required to comprehend long-term patterns and effects.

The purpose of this study consists of a systematic review of the different studies indexed in various scientific databases and publishers, including PubMed Central, IEEE Explore, Elsevier, Springer, the Multidisciplinary Digital Publishing Institute (MDPI), and the Association for Computing Machinery (ACM) published between January 2017 and October 2022, related to methods for the automation of the measurement of the Functional Reach Test variables and results with several sensors. This review is essential to understand the different usages of the FRT data and the techniques previously used in the literature to create a new methodology for the identification and registration of abnormal balance conditions continuously and remotely.

Compared to other literature reviews, the main contribution of this study is that it analyzes automated diagnosis algorithms and examines data acquisition, accessibility, and application. It focuses on using sensors and technological approaches for monitoring and measuring the Functional Reach Test, as well as on investigating the advantages of automatically calculating the Functional Reach Test results for different diseases.

The main conclusion is that more research studies need to be conducted on implementing technological methods to measure the Functional Reach Test results. The significant contribution of this review is to update the previously published review (Pires, Garcia & Zdravevski, 2020), verifying that most of the relevant studies were performed in 2021 and 2022. Finally, all the studies used at least statistical measurement, and the most used sensors are cameras, with the most studied disease being stroke recovery. The audience for which this review article is intended includes medical specialists in diseases that affect balance, including Parkinson’s disease and multiple sclerosis. It also seeks to clarify caregivers and physiotherapists who help older populations and society.

Survey Methodology

Research questions

These questions served as the foundation for this systematic review: (RQ1) Which sensors can be used to monitor various methods of measuring the Functional Reach Test? (RQ2) Which technological approaches can be utilized to measure the Functional Reach Test using sensory data? (RQ3) What are the advantages of automatically calculating the Functional Reach Test results for various diseases?

Inclusion criteria

Several inclusion criteria were used for this study, including the following: (1) studies that use sensors to measure the FRT parameters; (2) studies that present different FRT implementations; (3) studies that present the study’s purpose; (4) studies that clearly define the study’s population; (5) studies that present the results; (6) studies that present original research; (7) studies that were published between 2017 and 2022; and (8) studies written in English.

Exclusion criteria

Articles are also excluded based on the following criteria: (1) studies that did not report the use of technological equipment; (2) studies that did not provide details about the population characteristics; (3) studies that did not define or use technological methods for the analysis of the acquired data; (4) studies that were literature reviews or surveys; (5) studies that were study protocols or interviews; (6) studies that were not related to the healthcare subject with technological equipment’s.

Search strategy

The search was performed based on the PRISMA approach to obtain and analyze the Functional Reach Test studies published between January 2017 and October 2022. Initially, it was used as a framework for the automatic search in the following electronic databases: PubMed Central, IEEE Explore, Elsevier, Springer, MDPI, and ACM. The initial search phrases used to identify potentially relevant articles for this systematic review were “functional reach test with wearables” OR “functional reach test with mobile devices” OR “functional reach test”. Then, an NLP-based toolkit that provides a thorough search in these libraries using the supplied search terms automates the processing of the publications. Namely, the toolkit analyzes every identified article in the corresponding databases, automatically removing the duplicates and irrelevant ones based on the established criteria (Zdravevski et al., 2019). Therefore, the search was performed three times separately with these phrases in all digital libraries, and then, based on the identifiers, the software removed the duplicates. The selected studies were examined independently by the authors, who consensually decided on their relevance to this study. The studies were analyzed to determine the various approaches for automating the measurement of Functional Reach Test outcomes. The study was conducted on October 10, 2022.

Extraction of study characteristics

Several aspects were considered during the analysis of the articles. Table 1 shows the identified parameters in the studies, including the publication year, the study’s location, population, goal, methods employed, sensors and equipment used to gather the data, and diseases related to each study.

Table 1 Study analysis.

Study	Year of Publication	Location	Population	Purpose	Methods	Sensors/Equipment	Disease/Status	
Bruyneel et al. (2022)	2022	Switzerland; France	32 patients	Examine the reliability and validity of center of pressure (CoP) parameters measured during sitting balance on an unstable support, specifically for individuals in the subacute phase of stroke with hemiparesis.	Descriptive statistics; Shapiro–Wilk test; Bland-Altman analysis	Handheld dynamometer Dynamic sitting balance	Stroke	
Caimmi et al. (2022)	2022	Italy	19 participants	Effects of the intervention in reducing impairment in chronic stroke and to preliminarily verify the effects on activity	Descriptive statistics; Wilcoxon sign rank test; Linear regression; Pearson’s correlation; Evans’ classification	Robot Fully Assisted	Stroke	
Ghahramani, Rojas & Stirling (2022)	2022	Australia	38 participants	Explore different metrics that can potentially be used to identify early indications of balance loss and fall risk.	Mathematical models; Descriptive statistics; Pearson’s correlation	IMU sensors	Elderly	
Moriyama et al. (2022)	2022	Japan	21 young individuals, and 20 elderly individuals	Explore the relationship between the FRT value and the COPE and physical function in healthy young and older individuals classified according to movement patterns	Butterworth filter; Mathematical models; Descriptive statistics; Pearson’s correlation; Spearman’s correlation	Mation sensors Infrared cameras	Elderly	
Son, Muraki & Tochihara (2022)	2022	Japan; South Korea	9 subjects	Test methods for the investigation of the effect of personal protective equipment on mobility of firefighters.	Descriptive statistics; ANOVA; Student’s t-test; ANOVA; Tukey’s post-hoc test; Kruskal–Wallis test; Dunn’s pairwise tests	Motion sensors	N/D	
Ayed et al. (2021)	2021	Spain	19 participants	Investigate the feasibility and potential of using the Microsoft Kinect v2 sensor for measuring balance during the Functional Reach Test (FRT) to facilitate remote evaluation of patients.	Angle estimation; Mathematical models; Descriptive statistics; Pearson’s correlation; Student’s t-test; Shapiro–Wilk test	Microsoft Kinect v2 sensor	N/D	
Dewar et al. (2021)	2021	Australia	58 participants	Evaluate both forward and lateral FRT, for postural control in children with Cerebral Palsy	Descriptive statistics; Joint kinematic analysis; Student’s t-test; chi-square tests; Mann–Whitney U test; Spearman rank correlation coefficients	Force platforms Camera	Cerebral Palsy	
Marchesi et al. (2021)	2021	Italy	15 subjects	Investigate the upper-body kinematics and muscular activity during a modified version of the Functional Reach Test (FRT) in individuals who have experienced a stroke and are in the chronic phase of recovery	Mathematical models; Descriptive statistics; Pearson’s correlation; Spearman’s correlation; ANOVA; Anderson–Darling test; Mauchly’s test; Greenhouse–Geisser correction	Infrared cameras RGB cameras	Stroke	
Park, Son & Choi (2021)	2021	South Korea	16 participants	Clarify whether the distribution range of the forward reach distance and the relationship between the forward reach distance and the movement distance of the center of pressure differed depending on whether the controlled starting standing position during the functional reach test with an ankle joint strategy.	Descriptive statistics; Wilcoxon signed-rank test; Mann–Whitney U test	Force plate Cameras	N/D	
Nozu et al. (2021)	2021	Japan; United States of America	20 individuals	Characterize postural control strategies with and without disrupted somatosensory input during a dynamic balance task in people without chronic ankle sprain.	Butterworth filter; Mathematical models; Angle estimation; Descriptive statistics; Bonferroni correction; Shapiro–Wilk test;	Cameras Force plates	Ankle sprain	
Chen et al. (2020)	2020	Taiwan	35 individuals	Organize appropriate physical performance tests into a computerized early frailty screening platform, called frailty assessment tools (FAT) system, to detect individuals who are in the prefrail stage.	Angle estimation; Mathematical models; Descriptive statistics; Pearson’s correlation; Student’s t-test; Mann–Whitney U test	Motion sensors	Elderly	
Reguera-García et al. (2020)	2020	Spain	63 participants	Determine the evaluations in pressure mapping and verifying whether they are different between the three sample groups (multiple sclerosis, spinal cord injury and Friedreich’s ataxia), and to determine whether the variables extracted from the pressure mapping analysis are more sensitive than functional tests to evaluate the postural trunk control.	Descriptive statistics; Levene’s test; ANOVA; Bonferroni correction; Shapiro–Wilk test; Fisher statistic; Tukey post-hoc test; Kurskal-Wallis non-parametric H test; Pearson’s correlation; Spearman’s correlation; Linear regression	Pressure Imaging System	Multiple Sclerosis, Spinal Cord Injury; Friedreich’s Ataxia	
Santamaria et al. (2020)	2020	United States of America	4 subjects	Investigate the effectiveness of the robotic Trunk-Support-Trainer (TruST) in promoting functional and independent sitting in children with cerebral palsy (CP)	Angle estimation; Mathematical models; Descriptive statistics; ANOVA; ANCOVA	Robotic Trunk-Support-Trainer	Cerebral Palsy	
Fell et al. (2019)	2019	United States of America	35 participants	Assess the feasibility and effectiveness of using a combination of a mobile application and body-worn sensor technology for measuring functional outcomes in individual’s post-stroke.	Descriptive statistics; Spearman’s correlation	Mobile device NODE sensors	Stroke	
Fishbein et al. (2019)	2019	Israel	22 individuals	Investigate the feasibility of using a Virtual Reality-based dual task of an upper extremity while treadmill walking, to improve gait and functional balance performance of chronic poststroke survivors.	Descriptive statistics; ANOVA; Student’s t-test	SeeMe system	Stroke	
Tanaka et al. (2019)	2019	Japan	60 participants	Assess the accuracy of a markerless motion capture system in classifying the movement strategy during the Functional Reach Test	Angle estimation; Mathematical models; Descriptive statistics; Student’s t-test	Microsoft Kinect v2 sensor	N/D	
Verdini et al. (2019)	2019	Italy	48 individuals	Compare the NWBB [Nintendo Wii Balance Board] and a FP [force plates] to assess the error in the vertical force measure in two different highly dynamic tasks such as squatting (SQ) and functional reach test (FR).	Descriptive statistics; ANOVA; Student’s t-test; Mann–Whitney U test; Linear Regression	Nintendo Wii Balance Board Force plate	N/D	
Bao et al. (2018)	2018	United States of America	35 individuals	Assess the efficacy of long-term balance training with and without sensory augmentation among community-dwelling healthy older adults.	Descriptive statistics; Student’s t-test	Mobile device	Elderly	
Hsiao et al. (2018)	2018	Taiwan	442 participants	Investigate the dependability and associations between Kinect-derived measurements of forward reach distance and velocity, and their relationship with the conventional functional reach distance.	Descriptive statistics; Linear regression; Pearson’s correlation	Microsoft Kinect system	Elderly	
Mengarelli et al. (2018)	2018	Italy	48 subjects	Evaluate the validity of the Nintendo Wii Balance Board (NWBB) as a tool for measuring balance during the Functional Reach Test	Descriptive statistics; ANOVA; Student’s t-test; Shapiro–Wilk test; Mann–Whitney U test	Nintendo Wii Balance Board	Elderly	
De Luca et al. (2017)	2017	Italy	16 participants (9 female and 7 male)	Investigate the influence on sitting balance and paretic arm functions based on movements of the unimpaired arm	Descriptive statistics; Skillings–Mack test; Wilcoxon sign rank test; Bonferroni correction; Kolmogorov–Smirnov test	Exoskeleton Armeo Spring	Stroke	
Williams et al. (2017)	2017	United States of America	20 individuals	Develop a real-time system for assessing fall risk based on the Functional Reach Test	Angle estimation; Mathematical models; Descriptive statistics; Pearson’s correlation	Mobile device Wireless body sensors	Stroke	

Results

Figure 1 illustrates how the NLP tools automatically chose 22,312 research studies from the various databases. After deleting 1,178 duplicate studies discovered using the Digital Object Identifier (DOI), 21,134 publications out of the original 22,312 studies were retained. After an analysis of the article’s title with NLP, 8,039 publications were excluded as they did not fall into the inclusion criteria. After a similar analysis of the abstracts, 12,812 papers were additionally excluded. After the remaining studies were thoroughly analyzed, 261 research studies unrelated to the Functional Reach Test and using devices or simply literature reviews were eliminated. As a result, only 22 articles remained that are fully relevant to this study based on the inclusion and exclusion criteria.

Figure 1 Flow diagram of the selection of the relevant studies.

Multiple study characteristics were identified while analyzing the relevant studies. The query performed in this study retrieved papers published between January 2017 and October 2022. As reported in Table 1, four studies (18%) were published in 2022, six studies (27%) in 2021, three studies (14%) in 2020, four studies (18%) in 2019, three studies (14%) in 2018, and two studies (9%) in 2017. Even though none of the studies specified the use of standard devices or sensors, four studies (18%) used inertial sensors, five studies (23%) used force sensors, nine studies (41%) used imaging sensors, seven studies (32%) used robotic systems, four studies (18%) used mobile devices and other residual measuring devices were used in the different studies. Finally, regarding the diseases/status of the population analyzed in the studies, the participants had various conditions, including stroke analysis in seven studies (32%), elderly people analysis in six studies (27%), cerebral palsy analysis in two studies (9%), other conditions in two studies (9%), and population without specific diseases in five studies (23%). The following subsections summarize the findings of each study distributed by diseases/conditions, such as stroke, elderly, and other conditions.

Studies related to stroke

De Luca et al. (2017) proposed using a robot-assisted exoskeleton to help stroke survivors recover from their injuries. Sixteen chronic stroke survivors participated in 19 sessions of a training procedure. The exoskeleton was used with the unaffected arm to create an environment that promoted better spine alignment, enhanced intersegmental coordination, and decreased reliance on previously learned compensatory patterns. The authors claim this method improved postural control and motor skills in the unimpaired and untrained impaired arm.

Caimmi et al. (2022) assessed the viability and efficacy of a rehabilitation regimen using a robotic exoskeleton to help chronic stroke survivors lift their upper limbs against gravity. The study analyzes the safety and tolerability of the intervention and how it affects the 19 participants’ quality of life, spasticity, and motor performance. The results of this pilot study point to the feasibility and acceptability of the robot-assisted intervention, which has the potential to enhance upper-limb motor function and lessen stiffness in long-term stroke survivors. However, larger, randomized controlled trials must corroborate the pilot study’s findings.

Williams et al. (2017) created the mStroke system to measure the FRT and offer an unbiased evaluation of fall risk based on the estimated FRT scores. Based on several mobile health functions, mStroke is conceived, created, and distributed as an application (App) that runs on a hardware platform with an iPad and one or two wireless body motion sensors. The FRT reliability study was conducted on healthy adult subjects in a research setting with the necessary IRB approval, a wireless body sensor for mStroke, and an iPad to record the patient’s movement. The experimental findings validate the idea and viability of the mStroke FRT function, which the authors used to evaluate its dependability on two groups.

In their research, Fishbein et al. (2019) examined whether virtual reality-based dual-task training could benefit chronic poststroke survivors’ balance and walking more than conventional single-task training. Twenty-two participants were randomly allocated to either a standard single-task training group or a virtual reality-based dual-task training group for the preliminary study. The study’s findings suggested that virtual reality-based dual-task training could improve post-stroke patients’ walking and balance.

Center of pressure (COP) measurements from a force plate during an unstable sitting test were investigated by Bruyneel et al. (2022) to see if they might be used to assess trunk control in stroke survivors. Thirty-two stroke survivors participated in the study and received several assessments, including the Timed Up and Go test (TUG), the Modified Functional Reach Test (MFRT), the Balance Assessment in Sitting and Standing (BASSP), and isometric trunk strength. The results suggested that frontal and sagittal plane COP measures taken during an unstable sitting balance test on a seesaw could be considered legitimate metrics for assessing patients in the subacute stage of a stroke.

In a modified version of the Functional Reach Test (FRT), Marchesi et al. (2021) examined chronic stroke survivors’ upper body kinematics and muscular activity. The study’s goal, which involved 15 stroke survivors, was to learn how chronic stroke survivors perform this altered version of the FRT and how performance is related to upper body kinematics and muscular activation. To do this, a group of chronic stroke survivors underwent a modified FRT. At the same time, the study measured upper body kinematics (i.e., the movement of the upper body) and muscular activity (i.e., the activation of muscles). The results of this study can shed light on the biomechanical and neuromuscular mechanisms behind balance deficits in chronic stroke survivors, which could assist in guiding the creation of more successful rehabilitation plans for this population.

Fell et al. (2019) suggested utilizing the mStroke mobile health system. Two wearable sensors and a mobile application make up this system, which is intended to be a complete support management system for stroke survivors. The 35 post-stroke individuals in the study could comprehend and follow three-step instructions and walk 10 m without stopping or needing more than a little physical support, with or without an assistive device. The authors concluded that mStroke and other mobile applications are essential for long-term support and monitoring to minimize stroke recurrence and enhance recovery.

Studies related to older adults

To study the viability and validity of utilizing the Kinect system to measure dynamic balance and forward reach in older adults, Hsiao et al. (2018) examined the usage of the Kinect system by older adults. The functional reach test was completed by 442 individuals, all recorded by the Microsoft Kinect device. The findings point to the Kinect system’s potential utility as a low-cost, non-invasive technology for determining older adults’ balance and reach, which may be used to create individualized fitness interventions and fall prevention programs.

Mengarelli et al. (2018) looked at the reliability of using the Nintendo Wii Balance Board (WBB) as a tool to measure balance during the Functional Reach Test (FRT). The study aims to ascertain whether reliable balance measurements during the FRT can be obtained using the WBB. The study recruited 48 older adults as participants and compared the WBB measurements to those from a gold-standard force platform. By examining the agreement between the two sets of readings, the researchers evaluate the reliability of the WBB measurements.

Moriyama et al. (2022) examined the relationship between the FRT movement patterns and physical function in healthy young and older adults. The authors used tests that measured lower extremity muscle strength, balance, and gait speed to evaluate physical processes. The study included 41 participants—21 young and 20 older adults. The authors discovered that, in healthy individuals, movement patterns during the FRT were not a trustworthy predictor of physical function.

To determine whether the chest and pelvis coordination during the FRT may be used as a sign of balance deficit in older adults, Ghahramani, Rojas & Stirling (2022). Thirty-eight participants—27 older and 9 younger—participated in the study to see if monitoring the synchronization of the chest and pelvis during the test could reveal more about a person’s capacity for balance. The authors reasoned that older adults with balance issues would demonstrate less chest-to-pelvis coordination during the FRT.

Thirty-five older adults performed six physical performance tests, with a 3-minute break in between each test, as part of the work of Chen et al. (2020) intended to organize appropriate physical performance tests into a computerized early frailty screening platform called the frailty assessment tools (FAT) system. Data processing, recording, and frailty status classification were simultaneously carried out during the trials. The FAT program confirmed and saved the final test findings and the frailty group. According to the authors, the FAT scores of the participants in the older adults group can be used as a basis for therapeutic intervention techniques and to gauge the degree of frailty and rate of advancement.

The effectiveness of long-term balance training with and without sensory augmentation among healthy, community-dwelling older individuals was examined by Bao et al. (2018). Twelve individuals were divided into two equal groups for this study: control and experimental. They trained at home for eight weeks, engaging in three 45-minute workouts per week. In contrast to the control group, which did exercises without vibrotactile sensory augmentation, the experimental group used smartphone balance trainers for four of each exercise’s six repetitions. According to the authors, the study shows that using sensory augmentation devices for balance rehabilitation by community-dwelling, healthy older persons is feasible.

Studies related to other conditions

The validity of the Kids-Balance Evaluation Systems Test (Kids-BESTest) clinical criteria for the FRT forward and lateral against laboratory measures of postural control in children with cerebral palsy was examined by Dewar et al. (2021) The study involved 58 kids, ages 7 to 18, divided into 41 patients with usually developing conditions and 17 classified ambulant. For both study groups, the center-of-pressure (CoP) and joint mobility during reach were measured using kinematic markers during the forward and lateral FRT tests. The face, concurrent, and content validity tests were part of the data analysis. The authors claimed that the investigated tests, particularly the FRT forward test, are applicable.

The Trunk Support Trainer (TST), a robotic device, was assessed by Santamaria et al. (2020) to see how well it promoted active and independent sitting in kids with cerebral palsy (CP). The study looked at how the TST affected children with CP who have trouble sitting independently in terms of trunk control, sitting posture, and functional ability. In the trial with four kids, the TruST system assisted the child’s torso as needed while they practiced goal-oriented movements. The study shows that children with CP and postural sitting impairments can maximize their trunk and reaching control abilities with the TruST intervention.

Using pressure mapping technology, Reguera-García et al. (2020) assessed the postural control in seated patients with multiple sclerosis (MS), spinal cord injury (SCI), and Friedreich’s ataxia (FA). Ten adult patients with multiple sclerosis, spinal cord damage, and Friedreich’s ataxia participated in the case series study. Pressure mapping, the sitting Lateral Reach Test, the seated Functional Reach Test, the Berg Balance Scale, the Posture and Postural Ability Scale, the Function in Sitting Test, and the Trunk Control Test were the four functional assessments used. The study’s methodology is covered in the article, including the statistical analysis of the data and the use of pressure mapping technology (Pressure Imaging System X3Display) to detect pressure distribution and CoP displacement.

When identifying movement strategies during the FRT, Tanaka et al. (2019) compared the accuracy of a markerless motion capture system (MLS), which used Microsoft Kinect v2, to a marker-based motion capture system (MBS). The study’s 60 young, healthy participants were instructed to complete the FRT task of reaching forward while standing straight. Vicon was used as an MBS, and Microsoft Kinect v2 was used as an MLS to determine the coordinates of the hip, knee, and ankle joints. The authors hypothesized that the precise classification of movement patterns during the FRT utilizing markerless motion capture devices could have important ramifications for improving balance and fall risk assessment in older adults.

Nozu et al. (2021) investigated how healthy people’s postural control methods changed when their somatosensory input was disturbed during the Star Excursion Balance Test (SEBT). The authors proposed that wearing textured insoles would affect postural control methods during the SEBT, lowering performance in healthy individuals by obstructing somatosensory input. Twenty healthy participants participated in the study and completed the SEBT while having their center of pressure (COP) measured with and without the foam surface’s deception. According to the authors, the textured insoles disturbed somatosensory input, altering postural control methods and lowering performance during the SEBT in healthy subjects.

To assess balance and functional ability in clinical and research settings, Verdini et al. (2019) evaluated the precision of the Nintendo Wii Balance Board (NWBB) in measuring force during the squat and functional reach tests. The purpose of the study, which involved 48 young participants, was to evaluate the validity and reliability of the NWBB as a portable, affordable force measurement tool that may be useful for tracking and enhancing the rehabilitation of patients with balance disorders or other related problems.

Son, Muraki & Tochihara (2022) developed a test procedure to assess how firefighters’ mobility is affected by their use of personal protective equipment (PPE). In addition to an obstacle course, side-to-side leaps, a functional reach, a timed up-and-go, and range of motion tests, nine male firemen also underwent physical and functional balancing tests while wearing various PPE. As PPE weight increases, the wearer’s mobility decreases, as shown by decreased physical performance, functional balance capacity, and angular movement.

Park, Son & Choi (2021) looked into whether the controlled starting standing position (CSSP) during the functional reach test with an ankle joint strategy affected the distribution range of the forward reach distance (FoRD) and the relationship between the forward reach distance and the movement distance of the center of pressure (MSCOP). The experiments were carried out by sixteen healthy males, who held the pose for three seconds while photographed. The findings revealed a substantial association between the mean of the FoRD and the MDCOP and a narrowing of the FoRD’s distribution range with a CSSP.

Ayed et al. (2021) investigated a technique for remotely assessing patients’ balance using the Functional Reach Test (FRT) and the Microsoft Kinect v2 sensor. During the FRT, the study conducted by 19 healthy volunteers was captured utilizing the Microsoft Kinect v2 sensor. The goal was to measure the patient’s ability to balance accurately and reliably without requiring an in-person test. According to the authors, offering a remote evaluation of patients’ FRT may improve accessibility to balance assessment for people who might have limited mobility or difficult access to medical facilities.

Discussion

Interpretation of the results

In 2020, Pires, Garcia & Zdravevski (2020) performed a systematic review of studies published between 2009 and 2019, verifying that implementing the functional reach test using sensors could be helpful. Before the data acquisition with some individuals, the presented systematic review needed to be updated for the new knowledge in the literature, verifying that recent studies are being published, especially between 2019 and 2022. The result of this study shows that there is an emerging interest in this subject, highlighting its importance.

Based on Fig. 2, the studies were equally distributed from ten countries between 2017 and 2023. In 2017, two studies were equally distributed by the United States of America and Italy. In 2018, three studies were equally distributed by the United States of America, Italy, and Taiwan. In 2019, four studies were similarly distributed by Israel, Italy, Japan, and the United States of America. In 2020, three studies were equally distributed by Spain, Taiwan, and the United States of America. In 2021, four countries included authors from one study each, such as Australia, Italy, Spain, and the United States of America. The other two countries had authors from two studies, such as Japan and South Korea. Finally, in 2022, five studies were equally distributed from Australia, France, Italy, Japan, and Switzerland. In conclusion, the United States of America and Spain are the countries that most regularly perform studies on this subject.

Figure 2 The number of studies and their locations over the analyzed years.

Based on Fig. 3, the methods used in the different studies are the descriptive analysis in all studies, and the remaining methods are mathematical models and Pearson’s correlation in nine studies each, ANOVA in eight studies, angle estimation in six studies, Mann–Whitney U test and Shapiro Wilk test in five studies each, Spearman’s correlation in four studies, Bonferroni correction and Linear regression in three studies each, and Wilcoxon sign rank test and Butterworth filter in two studies each. Other less frequent methods are considered residual and used in different studies.

Figure 3 The number of studies and the most used methods.

Based on Fig. 4, the used devices/sensors in the different studies are the cameras in six studies, force plates, and motion sensors in four studies each, mobile devices and Microsoft Kinect v2 Sensor in three studies each, and Nintendo Wii Balance Boards in one study. Similarly, some less frequent methods are used in the different studies.

Figure 4 The number of studies and the corresponding most used sensors.

Based on Fig. 5, the different conditions analyzed in the studies are stroke in seven studies, older adults in six studies, cerebral palsy in two studies each, and the other five studies did not define the analyzed condition.

Figure 5 The number of studies and the most frequent conditions reported.

Based on Fig. 6, the authors verified that ten studies analyzed less than 20 individuals, six studies considered between 20 and 40 individuals, four studies recruited between 40 and 60 individuals, and only two evaluated more than 60 participants.

Figure 6 Number of studies and the distribution of the participant population.

Comparison of the different parameters

Figure 7 shows that descriptive statistics are the most widely used sets of methods applied in the different studies. The most significant relevance arises for measurements from force or motion sensors, emphasizing sensors from mobile devices, cameras, and video game devices.

Figure 7 Heatmap chart regarding the methods and sensors used in literature.

Similarly, it is observed in Fig. 8 that descriptive statistics is the most prominent method applied in the studies. The most significant relevance arises for conditions like aging or diseases like stroke. The authors do not identify several conditions, although they use descriptive statistics for their studies.

Figure 8 Heatmap chart regarding the methods and diseases used in the studies.

Comparison of the different studies

After analyzing the 22 selected studies, the benefits and limitations of each study are presented in Table 2. It is possible to see that the main limitations of the different studies are related to the sample size. However, other restrictions are verified, including the accuracy and calibration of the sensors that could lead to incorrect assessments of their risk of falling, the inexistence of standardization of the sensors’ positioning, the difficulty of integrating clinicians or therapists in the different studies, the complexity of the data acquired, the sensors can be intrusive or uncomfortable, other environmental factors such as temperature, humidity, or interference from other electronic devices, data privacy, the mobility impairment in the sample, and the complexity of the movements during the test. These are only examples that can be more comprehensively understood after reading the original studies for more information.

Table 2 Study’s overall benefits and limitations.

Study	Reason for selection	Results and Benefits	Limitations	
Bruyneel et al. (2022)	Detection of stability during the FRT with technological devices	The FRT shows valid parameters for assessing patients post-stroke, with reliability higher for center-of-pressure (CoP) length and velocity.	A control group is needed to understand brain injury’s impact on postural readjustments during unstable sitting balance and assess its predictive value for functional recovery.	
Caimmi et al. (2022)	Detection of stability during the FRT with technological devices	The authors introduced a rehabilitation intervention using robot-assisted functional movements against gravity. The intervention showed excellent compliance and reduced impairment in chronic stroke patients, even at 6-month follow-up. The results suggest that reduced impairment leads to improved activity, laying the groundwork for further studies to confirm findings and determine the optimal dose–response curve.	The study has several limitations, including a small sample size, dropouts, and the lack of a control group. It is unclear if patients improved better than other therapies, whether more intensive treatment led to better results, and if the assessor was not blinded. Further studies are needed to verify hypotheses on neuroplasticity, and the conclusions are based on clinical measures. The robot is not commercially available for rehabilitation, but a new one has been developed for further study.	
Ghahramani, Rojas & Stirling (2022)	Detection of stability during the FRT with technological devices	The study explores the impact of aging on motion strategy and chest and pelvis coordination during Functional Retraining (FRT). Results show decreased maximum angular rotation, dynamic time warping (DTW) cost, and coordination in older participants. The study suggests balance deficiency may increase fall risk in older individuals.	The study has limitations, including focusing solely on age and not considering other factors affecting balance and performance. It is still unclear what causes the more extensive mean continuous relative phase (CRP) in fallers. Future research should analyze the correlation between CRP and senility issues, categorize older participants based on fall history, and consider the imbalance in male and female participant groups. Further analysis of sex effects could be beneficial.	
Moriyama et al. (2022)	Detection of functional ability during the FRT with technological devices	The study found that the large hip strategy (LHS) can achieve high FRT values without an increased center of pressure (COP) forward movement, but this doesn’t necessarily indicate high physical function. The LHS may not reflect COPE or physical function, while the Small Hip Strategy (SHS) might reflect the center of pressure excursion (COPE) and some physical functions.	This study involved older individuals who participated in a health promotion project and could walk independently without a cane. The physical function assessment showed they were functional and at a low risk of falling. However, the findings were not generalizable to older individuals with high fall risk. The study also did not examine the effect of sex and movement strategies on fall risk and the relevance of these strategies in other assessments.	
Ayed et al. (2021)	Remote evaluation of the parameters related to FRT with technological devices	Two studies confirmed the linear correlation between Red Green Blue Depth (RGBD) devices and standard approaches for assessing balance tests. Corrected data showed no statistical differences between the two techniques. These studies are a first step towards implementing the system at home, with good repeatability and strong association with manual FRT, suggesting RGBD devices as a viable substitute for unsupervised balancing test measurement.	This study’s primary limitations are that the tests were carried out in a lab environment and that no patients were recruited for the research. It is also advised to conduct additional research before utilizing Red Green Blue Depth (RGBD) equipment to administer the test to selected individuals at home.	
Dewar et al. (2021)	Detection of stability during the FRT with technological devices	The study confirms the validity of forward FRT, with concurrent validity confirmed for lateral FRT. However, face and content validity was only confirmed when reaching the NP side in children with cerebral palsy (CP). Qualitative descriptors should be added to the Kids-BESTest FRT criteria.	Statistically significant differences were found between children with and without cerebral palsy (CP). Future studies may need samples to distinguish between postural techniques adopted by children with varying degrees of CP severity, motor types, distributions, and ages. Future research should focus on dynamic functional tasks, therapies to enhance stability limits, and postural methods to modulate sitting stability.	
Marchesi et al. (2021)	Detection of functional ability during the FRT with technological devices	The study examined trunk and contralesional arm muscles, finding asymmetric bilateral activations. Stroke survivors struggled to deactivate contralesional muscles, especially in the lateral direction. Repeated tasks improved reaching distance with increased trunk muscle activation. Reaching distance correlated negatively with time-up-and-go test scores.	This study focuses on the modified functional reach test in stroke survivors, focusing in two directions. The study’s small number of muscles limited understanding of complex coordinated patterns and synergic activations. Future research should include other muscles, trunk and leg muscles, and a more significant number of stroke survivors to understand differences due to impairment. The study also suggests that the modified functional reach test can be characterized quickly and paired with marker-less algorithms for movement analysis, which could be particularly useful during the pandemic.	
Park, Son & Choi (2021)	Detection of stability during the FRT with technological devices	The study found that posture control training in the sitting position using a virtual reality (VR) program was more effective in improving sitting balance and trunk stability in non-ambulatory children with cerebral palsy (CP), suggesting the development of future interventions to enhance independent movement performance and quality of life.	The study’s limitations include a small sample size, inconsistent effects due to short intervention duration, and insufficient consideration of balance and daily life activities. The Wii program is limited to children with cerebral palsy (CP), but the study has high clinical value. Further research is needed to develop a rehabilitation-specific program and long-term virtual reality (VR) training.	
Nozu et al. (2021)	Detection of functional ability during the FRT with technological devices	Posteromedial reach performance on unstable surfaces affects joint angles, COP, and COM, influenced by ankle movement strategies. Understanding somatosensory loss’s impact on dynamic balance tasks can help assess chronic ankle pathology.	The study has limitations, including the potential for foam pads to induce somatosensory disruption, altering ankle torque due to mechanical forces, and the need for ankle muscle electromyographic responses to understand movement control synergies. The sample size did not include people with pathology, and higher-end statistical models were not used to analyze all joints in a 3-dimensional motion-capture system. Additionally, the study cannot establish a minimally important change in movement control strategy for healthy people.	
Son, Muraki & Tochihara (2022)	Detection of functional ability during the FRT with technological devices	The study reveals that increasing personal protective equipment (PPE) weight decreases mobility, physical performance, and functional balance. Obstacle courses and step-ups are validated tests for measuring mobility, while functional balance tests, mainly Functional Reflex Tests (FRT), are practical. The findings can guide researchers, firefighters, and manufacturers in evaluating PPE mobility and developing comfortable, enhanced PPE.	The study’s statistical significance may be limited due to the participants’ lack of firefighting experience and the testing of firefighter personal protective equipment (PPE) in Japan. The test methods proposed may not apply to different races, countries, sexes, or PPE designs. Further validation, considering race, sex, and firefighter status, is required.	
Chen et al. (2020)	Detection of frailty syndrome based on FRT with technological equipment	The authors created a computerized platform for screening frailty syndrome tests, comparing prefrail and nonfrail groups. They developed a prefrail prediction equation, allowing for calculating FAT scores for elderly subjects, indicating frailty degree and progression rate.	The study has limitations due to the insufficient sample size of subjects and potential sampling bias. Four predictors of the prefrail stage were identified and used to develop the “prefrail prediction equation”, which allowed for calculating the FAT score to represent the degree of frailty of an elderly patient. The study also had a small age distribution, with only five subjects over 80 years old and one subject over 85 years old. The study suggests collaboration between experts from various areas is necessary to effectively assess and diagnose frailty syndrome, conduct swift screening and detection, and provide early intervention and treatment. Future studies should include more comprehensive physical function performance tests and elderly subjects to establish a more comprehensive frailty syndrome database.	
Reguera-García et al. (2020)	Detection of stability during the FRT with technological devices	Individuals with diverse neurological pathologies and similar functional tests show varying results in pressure mapping, indicating postural control issues. These individuals require specific treatment approaches, such as modifying wheelchair inclination or using orthopedic pillows, to reduce pressure ulcer risk.	The study’s main limitation is its small sample size, mainly involving people with severe disabilities and dependence in the same care center. Future research should analyze larger samples and patients with other neurological pathologies. It’s crucial to establish a correlation between sitting position and functional impairment to understand pressure mapping results. Future research should focus on specific physiotherapy protocols for people with multiple sclerosis, Friedreich’s ataxia, and spinal cord injuries.	
Santamaria et al. (2020)	Detection of functional ability during the FRT with technological devices	The study demonstrates the effectiveness of TruST-intervention in promoting functional and independent sitting in children with CP, GMFCS III-IV, and segmental trunk control impairments, demonstrating its potential to maximize trunk and reaching control abilities.	The therapeutic regimes for children were not discontinued, potentially preserving improvements during the 3-month washout period. A larger sample size is needed to test the generalization of postural and reaching control benefits to a similar cerebral palsy (CP) population. Randomized clinical trials are required to determine the superiority of TruST-intervention over matched-dose control therapies.	
Fell et al. (2019)	Detection of functional ability during the FRT with technological devices	The authors have found a strong correlation between clinician and mobile app scores for FRT, indicating the potential for a mobile health app to provide clinicians with objective movement data for recovery. This data could help identify potential health risks and trigger timely interventions, thus reducing complications. The findings encourage further development and testing of mobile health systems.	The study suggests future research on mStroke with a focus on caregiver collaboration and user-friendliness. It means a larger sample size and a focus on adherence support, streamlined communication between patients and healthcare teams, early identification and mitigation of complications, and caregiver support. The study also highlights the potential of machine learning in app development for the NIHSS Motor Arm and Motor Leg tests. The study emphasizes the need for further development and refinement of these apps.	
Fishbein et al. (2019)	Detection of stability during the FRT with technological devices	The study shows that Virtual Reality (VR)-based dual-task walking (DTW) can improve walking and balance in people after a stroke. It suggests combining different training sessions with multiple tasks using low-cost systems like SeeMe. VR systems also improve motivation and enjoyment during training, making them practical for community use. Smartphone technology could also be used to integrate tasks during treadmill activity. Further research is needed to determine the feasibility of these programs in different subpopulations and phases after stroke.	The study has limitations, including the lack of measurement of physiological changes or neurological mechanisms in the brain post-intervention. It suggests the need for future research to explore reactive postural control mechanisms and the significant heterogeneity in the damaged brain area, which could affect performance levels post-intervention.	
Tanaka et al. (2019)	Detection of functional ability during the FRT with technological devices	The markerless motion capture system (MLS) with Kinect system was used to measure the angular displacement of hip and ankle joints, suggesting its potential for accurately classifying movement strategies in FRT.	The study has several limitations, including a focus on ankle and hip joint angles, which may change if a hip angle method is adopted. However, the markerless motion capture system (MLS) data was valid, and the error between MLS and marker-based motion capture system (MBS) was smaller in ankle angles. The study did not examine the relationship between movement strategy and reach distance, suggesting future studies should use MLS with Kinect data.	
Verdini et al. (2019)	Detection of functional ability during the FRT with technological devices	The study demonstrates that the Nintendo Wii Balance Board can be effectively used for single-device measures during squat and functional reach tests, with high agreement with force plates (FP) data and low fixed biases. This low-cost device is particularly beneficial for research or clinical environments, as it accurately describes dynamic tasks like functional reach and squat tests, thereby promoting their use in research and clinical settings.	The percentage errors for center-of-pressure (COP)-parameters are consistent with those obtained without horizontal forces from force plates, indicating that the source of inaccuracy is device-dependent.	
Bao et al. (2018)	Detection of stability during the FRT with technological devices	A preliminary study found that vibrotactile sensory augmentation for eight weeks improved balance performance in healthy older adults. Participants showed increased visual and vestibular reliance, static and dynamic balance, and no pain, injuries, or falls. It supports sensory augmentation as a potential telerehabilitation tool.	The clinical outcome measures of gait-related activities show limited improvements due to the absence of significant sensory augmentation during gait exercises and limited transfer effects from standing exercises.	
Hsiao et al. (2018)	Detection of stability during the FRT with technological devices	The study suggests that the Kinect system can be a reliable alternative to traditional functional reach tests for assessing balance in older adults. It proposes two parameters of forward reach: maximum distance and maximal velocity at the halfway time. The results show a strong correlation between these parameters and traditional functional reach, measuring postural stability and static balance. However, the older group had lower maximal forward reach than the younger group. The Kinect system is simple, inexpensive, repeatable, and age-sensitive, and walking speed and grip strength are correlated with maximal velocity at the halfway time.	The study has limitations, including an oversimplified model for balance components, a focus on single-axis movement, a single facility for elderly persons, and arbitrary comparator outcome measures. Balance is a complex process that requires coordination and constant adjustment. Factors like acceleration and deceleration must be considered when evaluating an individual’s balance function. The study also lacks validity and reliability for using the velocity of forward reach as a dynamic balance function indicator. Further testing is needed for generalization.	
Mengarelli et al. (2018)	Detection of functional ability during the FRT with technological devices	The study evaluates the performance of the Nintendo Balance Board (NBB) in assessing balance measures in Functional Regression Testing (FRT). NBB has high consistency and linearity, making it suitable for quantifying center-of-pressure (COP) trajectories and related measures. Noise affects all COP parameters, suggesting its use in various loads for FRT evaluations. The findings could enhance low-cost instrumentation for balance assessment, especially in medical environments like rehabilitation centers. NBB’s portability, affordability, and lack of dedicated staff suit home-focused scenarios.	Further research is needed to assess the inter-device variability and test-retest reproducibility of the Nintendo Balance Board (NBB) for full usability in FRT and clinical relevance evaluations.	
De Luca et al. (2017)	Detection of stability during the FRT with technological devices	The study emphasizes the significance of considering all body schema in a rehabilitation robotic program rather than concentrating solely on the impaired side.	The study did not investigate treatment effects on a control group, and the therapists involved in the training were not blinded to the experiment’s goal, allowing them to help stroke survivors achieve established goals. The authors found similar motor synergies between two arms due to three factors: Armeo’s gravity support, restricted training to robotic joints, and a larger population of mildly impaired subjects. Changes in these factors may result in different results.	
Williams et al. (2017)	Detection of stability during the FRT with technological devices	The authors developed a mobile health system called mStroke, which performs fall risk assessment using FRT. Three reach distance measures were tested on healthy adult subjects, confirming the system’s concept and feasibility.	Experimental results show promising results, but room for improvement is suggested for FRT function in mStroke. More motion sensors, like those on the shoulder or arm, could capture more detailed body movements.	

Limitations

The review discusses several limitations in the studies analyzed, including small sample sizes, sensor accuracy, and calibration, lack of standardization in sensor positioning, difficulty in integrating clinicians or therapists, complexity of data, intrusiveness of sensors, environmental factors, data privacy, mobility impairment in samples, and complexity of movements during the FRT. These limitations can impact the generalizability and robustness of the findings, as well as the accuracy and reliability of the data. Sensor technology inaccuracies could lead to incorrect risk assessments, especially in fall risk evaluation. The lack of standardization in sensor positioning can also affect data consistency and reliability. Integrating clinicians or therapists into the studies could also impact the practical application of the research findings. Ensuring data privacy is a significant concern.

Research directions

The article outlines research directions to address gaps in sensor use for Functional Reach Test. This test might use sensors to measure distance, but key areas include confirming sensor accuracy and reliability, integrating and testing new sensor models, and enhancing sensor technology for clinical and research settings. Additionally, the article emphasizes the need for continuous research in data analysis and interpretation due to the complexity and vast amount of data collected during the test. It will help ensure reliable data for assessing functional reach and balance, enhancing the sensors’ effectiveness in various settings. Clinical applications and longitudinal studies are also needed to understand long-term trends and outcomes.

Final remarks

After reviewing the 22 relevant studies, we summarize the main findings of this systematic review as follows. Considering the RQ1 (“Which sensors can be used to monitor various methods of measuring the Functional Reach Test?”), at first glance, we concluded that camera-based devices are preferred. Next, they are complemented by motion-based sensors, such as those embedded in mobile devices or the specific ones from video game platforms. The way these sensors work and their versatility pave the way for different solutions and approaches for a wide range of studies and measurement protocols.

Regarding RQ2 (“Which technological approaches can be utilized to measure the Functional Reach Test using the sensor data?”), we conclude that the most common approach is to use statistical models to extract meaningful information from the data gathered from the sensors. The statistical models reported by the authors vary a lot, partly due to the wide range of sensors used. Usually, authors choose methods that best adapt to each sensor’s data type so a wide range of methods can be used.

Finally, for RQ3 (“What are the advantages of automatically calculating the Functional Reach Test results for various diseases?”) we identified the main advantage the possibility of promoting autonomy to users/patients who, with the use of different equipment, can perform the FRT and obtain results in real-time. As a final comment, there are some promising investigations in this subject, although these are still embryonic, and to the best of our knowledge, there is no solution on the market.

Conclusions

This article has systematically reviewed multiple sensor systems for measuring Functional Reach Test. A total of 22 studies were considered relevant based on the inclusion criteria, meaning this is an emerging research area, considering that 13 studies were published after the previous review from two years ago. This means that the subject is starting to be more attractive for researchers and medical practitioners, and there is an urgent need to develop an automated system for measuring the FRT.

Sensors are used in the FRT to assess distance, yet there are still unanswered research questions. These consist of longitudinal research, data analysis method development, sensor technology developments, clinical applications, and validation of the accuracy and dependability of sensors. Further research is required to integrate and test novel sensor types or models, confirm the accuracy and dependability of sensors in various populations and contexts, and comprehend long-term patterns and consequences. Further research is required to optimize the use of sensors in clinical practice and improve patient outcomes.

In future work, the authors intend to create a dataset with Portuguese older adults performing the FRT to be measured with the mobile devices’ sensors. After that, a measuring method will be developed to create a system for measuring physical conditions based on the FRT, allowing remote monitoring and estimating the health status.

Additional Information and Declarations

Competing Interests

Author Contributions

Data Availability

Ivan Miguel Pires, Paulo Jorge Coelho, Eftim Zdravevski and Carlos Albuquerque are Academic Editors for PeerJ.

Luís Francisco conceived and designed the experiments, performed the experiments, analyzed the data, performed the computation work, prepared figures and/or tables, authored or reviewed drafts of the article, and approved the final draft.

João Duarte conceived and designed the experiments, performed the experiments, analyzed the data, performed the computation work, prepared figures and/or tables, authored or reviewed drafts of the article, and approved the final draft.

António Nunes Godinho conceived and designed the experiments, performed the experiments, analyzed the data, performed the computation work, prepared figures and/or tables, authored or reviewed drafts of the article, and approved the final draft.

Eftim Zdravevski conceived and designed the experiments, performed the experiments, analyzed the data, performed the computation work, prepared figures and/or tables, authored or reviewed drafts of the article, and approved the final draft.

Carlos Albuquerque conceived and designed the experiments, performed the experiments, analyzed the data, performed the computation work, prepared figures and/or tables, authored or reviewed drafts of the article, and approved the final draft.

Ivan Miguel Pires conceived and designed the experiments, performed the experiments, analyzed the data, performed the computation work, prepared figures and/or tables, authored or reviewed drafts of the article, and approved the final draft.

Paulo Jorge Coelho conceived and designed the experiments, performed the experiments, analyzed the data, performed the computation work, prepared figures and/or tables, authored or reviewed drafts of the article, and approved the final draft.

The following information was supplied regarding data availability:

This is a systematic review.

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
