# Peer review of "Sensor-based systems for the measurement of Functional Reach Test results: a systematic review"

_PeerJ Computer Science, doi:10.7717/peerj-cs.1823_

## Round 0.1 · original submission · Major Revisions

Dear authors, you are advised to critically respond to all comments point by point, while preparing for the response.

·

Basic reporting

The paper comprises several papers pertaining to the Functional Reach Test, comprehensively addressing various facets of the subject area. While the literature is well-presented, the criteria for selecting the final papers are not explicitly elucidated. Nonetheless, this paper holds merit as a valuable addition to the journal, albeit necessitating some modifications.

Experimental design

The papers align with the journal's Aims and Scope. Proper citation of resources is evident, and the chosen papers are succinctly and clearly summarized. Furthermore, the review is thoughtfully structured and presents information in an easily comprehensible manner.

Validity of the findings

The finding of the paper is on point, and can lead future researchers to have a broader view of the topic. The paper achieves this by comparing multiple approaches and showing what the selected papers have in common.

Additional comments

The paper is well structured and the goal is clear. However, since the main goal of the authors is to show statistics related to papers that study the FRT results, the images must be better. In fact, all the proposed images have a very low quality. This is the main issue, and must be addressed. Moreover, due to the nature of the paper, I believe that a reason on why each paper was selected must be included.

Reviewer 2 ·

Basic reporting

This manuscript endeavors to provide a literature review on sensor-based systems for measuring Functional Reach Test (FRT) results across diverse populations and conditions. The attempt to synthesize existing literature on this topic is commendable. Below are some areas of improvement and suggestions to enhance the clarity, coherence, and depth of the manuscript:

1. The introduction could benefit from a more detailed exposition of the research problem, the existing gap in the literature, and the research questions the review intends to address. Clearly articulating the purpose, scope, and objectives of the review, along with a justification for the chosen topic and methods, will provide readers with a solid foundation for understanding the review's significance.

2. It appears that Figures 1, 7, and 8 have a lower resolution which may hinder a clear understanding of the content. Improving the resolution and clarity of these figures would greatly enhance the visual representation of the findings.

3. There's a minor inconsistency in Line 122 regarding the number of studies reviewed. Rectifying this inconsistency will improve the accuracy and coherence of the manuscript.

Experimental design

1. While the manuscript presents a range of studies, a deeper level of analysis, synthesis, and discussion could provide more valuable insights. Ensuring that the conclusions are well-supported by the evidence and arguments presented in the review will enhance the credibility and contribution of the manuscript.

2. The results section is well-intentioned but may benefit from a more structured approach. Incorporating subtitles could aid in organizing the content more logically, and synthesizing or comparing the findings of the reviewed studies could offer more meaningful insights.

Validity of the findings

1. The results section is well-intentioned but may benefit from a more structured approach. Incorporating subtitles could aid in organizing the content more logically, and synthesizing or comparing the findings of the reviewed studies could offer more meaningful insights.

2. The discussion section could delve deeper into how the findings address the research questions and contribute to the existing body of knowledge. Discussing the implications and applications of the results, alongside suggesting directions for future research, will enrich the manuscript. Acknowledging the limitations of the review and exploring the reasons behind the preference for certain sensors in more depth could provide a more comprehensive understanding.

Reviewer 3 ·

Basic reporting

The overall contents of the paper are satisfactory; however, the author might consider reorganizing the information in a more coherent and visually appealing way. The abstract should be enhanced by including the findings of the present study. The paper is written at a proficient level of English and demonstrates a comprehensive comprehension of the explored topic.

Experimental design

The study technique session would benefit from enhancements such as the inclusion of a diagram and a comprehensive explanation of how the chosen technique differs from others. Additionally, it would be valuable to elaborate on the anticipated results of the study. In general, the document contains an adequate amount of material for the intended audience. The survey methodology should be thoroughly discussed, including its influence and results. The sources have been appropriately referenced. The organization of the paper could be improved.

Validity of the findings

The conclusion and discussion sections could be enhanced by addressing existing gaps and suggesting potential avenues for further research. An argument that is deemed acceptable in order to fulfill the objectives of a research study.

The length of the written conclusions is insufficient and should be expanded to provide a comprehensive explanation, including a discussion of future directions.

Additional comments

Improve the Abstract, Methodology, Discussion, and Conclusion sections.

---

## Round 0.2 · Minor Revisions

One of the reviewers would like to see further improvements over a few issues. Please address these issues.

·

Basic reporting

no comment

Experimental design

no comment

Validity of the findings

no comment

Additional comments

The authors have revised as per the comments.

Reviewer 2 ·

Basic reporting

I appreciate the effort put into the revised version of the manuscript. However, I still have concerns regarding several aspects of the paper that require further revisions to enhance its quality and clarity.

The manuscript would benefit from meticulous proofreading to correct grammar errors and enhance language clarity. Some sections of the manuscript are somewhat difficult to comprehend. Engaging a fluent English speaker to refine the manuscript could help eliminate such errors and improve the overall readability. For instance:
"In older persons and those with neurological diseases like Parkinsonís disease or multiple sclerosis, balance problems are frequently an issue"
"The two are closely related since sensors are included in mobile devices to provide various functions and capabilities."

The issue with the quality of figures persists in the revised version. The text in these figures remains unclear due to low resolution, making it difficult for readers to discern the content. Improving the resolution and ensuring that the text is legible is essential for effective visual representation. Also, it is beneficial to ensure that figures also contribute effectively to the understanding of the content.

Experimental design

The manuscript repeats the exact same research questions in the introduction and the methodology without modification or rephrasing. To enhance the clarity and coherence of the paper, consider refining or rephrasing the research questions to avoid redundancy and to provide a fresh perspective on the study.

Validity of the findings

The discussion and conclusion sections of the manuscript lack depth. It is crucial to delve deeper into the findings, addressing their implications and significance more comprehensively. Additionally, explore existing gaps in the literature and suggest potential avenues for further research in these sections.

Reviewer 3 ·

Basic reporting

The author has revised the script and good to be accepted now.

Experimental design

Method and work flow is explained properly. Literature review is in detailed.

Validity of the findings

Author has added the comparison table and explained properly.

Additional comments

Recheck the submitted word file template.

---

## Round 0.3 · accepted · Accept

The authors have fully addressed the issues raised by the reviewers. I now recommend accepting this paper.